# A Plasma Extracellular Vesicle-Derived microRNA Signature as a Potential Biomarker for Subclinical Coronary Atherosclerosis

**DOI:** 10.3390/ijms26178727

**Published:** 2025-09-07

**Authors:** Mario Peña-Peña, Óscar Zepeda-García, Rosalinda Posadas-Sánchez, Fausto Sánchez-Muñoz, Mayra Domínguez-Pérez, Juan Alfonso Martínez-Greene, Fabiola López-Bautista, Adrián Hernández-Díazcouder, Rogelio F. Jiménez-Ortega, Alejandra Idan Valencia-Cruz, Adrián Nuñez-Salgado, Isaac Emanuel Mani-Arellano, Karina Martínez-Flores, Teresa Villarreal-Molina, Eduardo Martínez-Martínez, Leonor Jacobo-Albavera

**Affiliations:** 1Sección de Estudios de Posgrado, Escuela Superior de Medicina, Instituto Politécnico Nacional, Mexico City 11340, Mexico; marionutricion2017@gmail.com; 2Departamento de Fisiología, Instituto Nacional de Cardiología Ignacio Chávez, Mexico City 14080, Mexico; fausto22@yahoo.com; 3Laboratorio de Genómica de Enfermedades Cardiovasculares, Instituto Nacional de Medicina Genómica (INMEGEN), Mexico City 14610, Mexico; oscar.zega@gmail.com (Ó.Z.-G.); mdominguez@inmegen.gob.mx (M.D.-P.); adrianuez53@gmail.com (A.N.-S.); isaac2019.unam@gmail.com (I.E.M.-A.); mvillareal@inmegen.gob.mx (T.V.-M.); 4Departmento de Endocrinología, Instituto Nacional de Cardiología Ignacio Chávez, Mexico City 14080, Mexico; rossy_posadas_s@yahoo.it (R.P.-S.); nutrifabs@gmail.com (F.L.-B.); 5Laboratorio de Comunicación Celular y Vesículas Extracelulares, Instituto Nacional de Medicina Genómica (INMEGEN), Mexico City 14610, Mexico; martinezgreene@ie-freiburg.mpg.de; 6Unidad de Investigación Médica en Bioquímica, Hospital de Especialidades, Centro Médico Nacional Siglo XXI, Instituto Mexicano del Seguro Social, Mexico City 06720, Mexico; adrian.hernandez.diazc@gmail.com; 7Servicio de Medicina Genómica, Instituto Nacional de Rehabilitación Luis Guillermo Ibarra Ibarra, Mexico City 14389, Mexico; rfjimenez@inr.gob.mx; 8Unidad de Proteómica, Instituto Nacional de Medicina Genómica (INMEGEN), Mexico City 14610, Mexico; aivalencia@inmegen.gob.mx; 9Laboratorio de Líquido Sinovial, Instituto Nacional de Rehabilitación Luis Guillermo Ibarra Ibarra, Mexico City 14389, Mexico; karinabiologist@hotmail.com

**Keywords:** extracellular vesicles, microRNAs, subclinical coronary atherosclerosis, biomarker discovery, bioinformatics

## Abstract

Subclinical coronary atherosclerosis (SCA) is an early stage of coronary artery disease (CAD) that often goes unrecognized until clinical events occur. Identifying circulating molecular biomarkers could improve early diagnosis and risk assessment in asymptomatic individuals. This study employed a two-phase approach to identify plasma extracellular vesicle (EV)-derived microRNAs (miRNAs) associated with SCA. In the discovery phase, plasma samples from male participants were analyzed using Affymetrix GeneChip miRNA 4.0 microarrays. Differentially expressed miRNAs were refined through bioinformatic analysis, cross-species comparison with murine data, and target gene prediction. In the validation phase, six candidate miRNAs were quantified by RT-qPCR in an independent cohort. Six miRNAs were differentially expressed between individuals with SCA and controls. Among these, the combination of miR-146b-5p, miR-4701-3p, and miR-1180-3p demonstrated a high discriminative capacity for SCA (AUC = 0.8281; sensitivity = 93.75%; specificity = 93.75%). Functional enrichment analysis revealed that predicted target genes are involved in key atherosclerosis-related pathways, including inflammation, lipid metabolism, and vascular remodeling. EV-derived miRNAs may serve as non-invasive biomarkers for the early detection of coronary atherosclerosis. These findings provide insight into the molecular processes underlying subclinical vascular disease and support the integration of EV-associated miRNAs into preventive cardiology strategies.

## 1. Introduction

Atherosclerosis is a major contributor to cardiovascular diseases (CVD), leading annually to approximately 17.9 million deaths worldwide [1]. This chronic inflammatory disease is characterized by lesions within the intimal layer of blood vessels, progressively resulting in the formation of atherosclerotic plaques [2]. Such plaques form through complex interactions involving endothelial activation, accumulation of lipoproteins, and infiltration of inflammatory cells. As plaques mature, they can progressively obstruct blood flow or rupture, causing severe ischemic complications such as myocardial infarction, stroke, and renal failure [3].

Given the significant health burden posed by CVD, the American Heart Association (AHA) in 2010 established seven metrics designed to assess and monitor cardiovascular health, emphasizing lifestyle modifications including maintaining a healthy diet, regular physical activity, tobacco cessation, as well as pharmacological interventions such as statins for lipid management [4]. However, despite these guidelines, achieving optimal lipid control remains challenging, even with high-intensity lipid-lowering therapies [5]. Consequently, additional therapeutic options, such as cholesterol-binding agents like ezetimibe and inhibitors of proprotein convertase subtilisin/kexin type 9 (PCSK9), have emerged [6,7]. In severe cases of atherosclerosis, surgical interventions or stent-based therapies are often necessary [8].

While advanced atherosclerosis leads to clinically apparent symptoms, initial stages frequently involve asymptomatic lesions known as SCA [9]. Despite their silent nature, these early lesions carry substantial risk due to their potential progression to acute myocardial infarction, a primary cause of morbidity and mortality globally and particularly prevalent in Mexico [10,11,12].

To detect and monitor atherosclerosis, current diagnostic approaches largely depend on traditional biomarkers such as total cholesterol, low-density lipoprotein cholesterol (LDL-C), serum triglycerides, and inflammatory markers like C-reactive protein [13]. Although these biomarkers are clinically useful, they have critical limitations, including inadequate sensitivity and specificity for reliably predicting severe cardiovascular events [14]. Additionally, many diagnostic techniques used to evaluate atherosclerosis involve invasive methods, which pose procedural risks, incur high costs, and offer limited predictive capability [15]. These constraints underscore an urgent need for novel, minimally invasive biomarkers capable of accurately detecting and stratifying risk in patients with SCA [16].

Although the absence of traditional cardiovascular risk factors (CVRFs) is typically associated with a low probability of developing atherosclerosis, clinical events still occur in individuals classified as low-risk under current definitions [17]. These criteria often include untreated blood pressure < 140/90 mmHg, fasting glucose < 126 mg/dL, total cholesterol < 240 mg/dL, LDL-C < 160 mg/dL, and high-density lipoprotein cholesterol (HDL-C) ≥ 40 mg/dL [13,14]. The persistence of cardiovascular events in such individuals suggests that conventional risk stratification may underestimate subclinical disease burden, reinforcing the need for additional biomarkers to improve early detection [18].

Recent studies have highlighted the biological relevance of EVs in CVD [19,20,21]. These are small, membrane-bound particles secreted by various cell types that participate in intercellular communication under both physiological and pathological conditions [22]. EVs carry a diverse molecular cargo, including proteins, lipids, and nucleic acids, which reflects the status of their cells of origin [23]. Among their most studied functions in recent years is their ability to transport microRNAs (miRNAs) [24,25].

miRNAs are small non-coding RNA molecules, approximately 21 to 23 nucleotides in length, that post-transcriptionally regulate gene expression [26,27]. They are involved in fundamental cellular processes such as proliferation, differentiation, metabolism, apoptosis, and inflammation [28]. The stability of miRNAs in circulation and their selective enrichment within EVs support their utility as minimally invasive biomarkers in various disease contexts, including SCA [29].

Recent studies indicate that EV-associated miRNAs have significant potential as reliable biomarkers for the early detection and monitoring of SCA [30,31,32]. Consequently, exploring EV-derived miRNAs presents an essential opportunity to enhance diagnostic precision, improve risk assessment, and identify novel therapeutic targets for more effectively managing and potentially preventing atherosclerosis [29,33]. Therefore, the aim of this study was to identify a plasma EV-derived miRNA signature associated with SCA as a potential biomarker.

## 2. Results

### 2.1. Discovery Phase

#### 2.1.1. Main Clinical Features of Study Participants

Table 3 shows the main characteristics of SCA cases and controls. No significant differences in age, anthropometric, or biochemical parameters between groups were observed (*p* > 0.5). Only the coronary artery calcium score (CACS) showed significant differences according to the study design.

#### 2.1.2. Characterization of EVs

TEM and WB images of plasma EVs are shown in Figure 1. The characteristic morphology and size of vesicles are observed in Figure 1a. Total protein concentration in EV preparations showed no significant differences between groups (Figure 1b). EV-enriched markers CD63 and CD9 were observed by WB in all samples (Figure 1c), and densitometry revealed no significant differences in the intensity of CD63 and CD9 signals between study groups (Figure 1d and Figure 1e, respectively). These results confirm the successful isolation and characterization of EVs.

#### 2.1.3. Identification of Candidate miRNAs and Target Genes

Microarray analysis identified 24 DE-miRNAs between individuals with SCA and controls (−log2FC ≤ −0.5 or ≥0.5, *p* < 0.05), including 13 downregulated and 11 upregulated miRNAs (Figure 2a, Appendix A). To elucidate whether the DE-miRNAs identified in human samples were also differentially expressed in an advanced stage of the disease, we performed data mining of total RNA expression profiles from an established murine model of atherosclerosis (GSE137581). In this dataset, 113 DE-miRNAs were identified between atherosclerotic and control mice, with 47 downregulated and 67 upregulated (Figure 2b, Appendix A). Six miRNAs were found to be differentially expressed in both models (Figure 2c). Figure 2d shows the expression profiles of these shared miRNAs in human samples, where upregulated miRNAs are represented in red and downregulated in green.

Forty target genes for hsa-miR-487b-3p, 189 for hsa-miR-379-5p, 66 for hsa-miR-146b-5p, 360 for hsa-miR-4701-3p, 193 for hsa-miR-6849-5p and 32 target genes for hsa-miR-1180-3p were identified by at least 3 databases (Figure 3a, Appendix A). Moreover, data from the Tampere vascular study identified 791 differentially expressed genes (DEGs), with 307 downregulated and 484 upregulated (Figure 3b, Appendix A). Thirty-six genes were identified, both as target genes for DE-miRNAs and DEG in atherosclerotic plaques (Figure 3c).

#### 2.1.4. Pathway Enrichment Analysis and Interaction Networks

Figure 4a shows the 16 signaling pathways together with the number of genes and the -log10(FDR) values associated with each. The analysis of selected target genes revealed 16 enriched pathways, some related to carbohydrate and amino acid metabolism (Figure 4a,b). The interaction network between miRNAs and their potential target genes is shown in Figure 4c. This network suggests that miRNAs play an important regulatory role in the progression of atherosclerosis.

### 2.2. Validation Phase

#### 2.2.1. Main Clinical Features of Study Participants

In the validation cohort, median age was similar in cases and controls (67.0 years [57.0–72.5] vs. 67.0 years [61.0–72.0]; *p* = 0.609). Moreover, no statistically significant differences between groups were found in anthropometric or biochemical parameters. Only CACS values differed significantly according to the study design (Table 4).

#### 2.2.2. Validation of Differentially Expressed miRNAs by RT-qPCR

The six miRNAs that were differential expressed in both models (ACS vs. controls, and atherosclerotic mice model; Figure 2c) were measured in plasma-derived EVs of validation cohort participants using RT-qPCR. All showed statistically significant differences between groups (Table 1; *p* < 0.05).

#### 2.2.3. Identification of a miRNA Signature Through ROC Curve Analysis

Receiver operating characteristic (ROC) curve analysis was used to explore the diagnostic potential of the differentially expressed miRNAs. Individual ROC curves were generated for each of the six miRNAs. AUC values were calculated to assess their discriminatory capacity. Additionally, a multivariable logistic regression model was applied to evaluate combinations of these miRNAs, aiming to identify an optimal plasma-based signature with enhanced sensitivity and specificity for detecting SCA (Table 2).

Assessed individually, miR-146b-5p presented an AUC of 0.7543, miR-4701-3p an AUC of 0.7404, and miR-1180-3p an AUC of 0.686, supporting their relevance within the identified signature. To further improve diagnostic accuracy, we evaluated combinations of miRNAs using multivariable logistic regression followed by ROC curve analysis. Among the different combinations tested, the signature with the best clinical performance consisted of miR-146b-5p, miR-4701-3p, and miR-1180-3p, which achieved an AUC of 0.8281, with 93.75% sensitivity and 93.75% specificity, indicating strong discriminatory capacity (Figure 5).

## 3. Discussion

This study identified six miRNAs (miR-146b-5p, miR-4701-3p, miR-487b-3p, miR-379-5p, miR-1180-3p, and miR-6849-5p) differentially expressed between individuals with SCA and controls. Among these, a combination of three miRNAs (miR-146b-5p, miR-4701-3p, and miR-1180-3p) demonstrated the highest discriminative capacity, achieving an AUC of 0.8281 with 93.75% sensitivity and 93.75% specificity. We propose this three-miRNA signature, established through binary logistic regression and validated via ROC curve analysis, as a potential diagnostic signature for SCA. Our results highlight the potential clinical relevance of plasma EV-derived miRNAs as minimally invasive biomarkers for the early identification of SCA in asymptomatic individuals.

Importantly, the three miRNAs comprising the diagnostic signature (miR-146b-5p, miR-4701-3p, and miR-1180-3p) are involved in key atherogenic processes, including inflammation, endothelial dysfunction, angiogenesis, and thrombosis. miR-146b-5p is known to regulate the NF-κB pathway, monocyte adhesion, and endothelial barrier integrity, suggesting a role in modulating vascular inflammation and leukocyte-endothelium interactions [34,35,36,37,38,39]. miR-4701-3p, although less extensively studied, has been associated with angiogenic activation via TOB2 (Transducer of ERBB2, 2) suppression and implicated in endothelial cell proliferation and platelet reactivity [40,41,42]. Similarly, miR-1180-3p contributes to vascular remodeling by inhibiting profibrotic signaling via E26 transformation-specific sequence 1 (ETS1) and has shown inverse associations with platelet activation markers in CAD patients [43,44,45]. Notably, while miR-146b-5p and miR-1180-3p were downregulated in SCA, miR-4701-3p was upregulated, suggesting that the signature captures both suppressed protective responses and compensatory activation mechanisms. These complementary expression patterns and biological functions may enhance the overall diagnostic capacity of the miRNA signature by reflecting multiple facets of subclinical atherogenesis.

In addition to the signature, three other EV-derived miRNAs were significantly altered in individuals with SCA. miR-487b-3p, a member of the 14q32 cluster, has been linked to vascular remodeling and post-ischemic neovascularization, and its downregulation may reflect adaptive mechanisms to preserve endothelial repair capacity. miR-379-5p was found to be upregulated and has been associated with early vascular alterations through its interaction with IL-27 polymorphisms, showing functional relevance in asymptomatic coronary atherosclerosis within the Mexican population [46]. Finally, miR-6849-5p remains poorly characterized, though preliminary data from transcriptomic studies suggest a possible involvement in vascular dysfunction. While further studies are required to elucidate its role, its differential expression reinforces the exploratory potential of EV-miRNAs in uncovering novel regulatory elements in early atherosclerosis.

The interaction network between the six miRNAs analyzed and their potential target genes suggests that these miRNAs play an important regulatory role in the progression of atherosclerosis. Upregulated genes, such as *WAS*, *SYK*, *PIK3CD*, *LIPA* and *OAS2*, are related to inflammation and lipid metabolism [47]. Downregulated genes, such as *KCNK17*, *PRKG2* and *MGST1*, are involved in vascular protective functions and antioxidant signaling [48].

More than 100 publications analyze the role of circulating miRNAs as atherosclerosis biomarkers in humans (clinical and experimental), animal models, and in vitro. However, less than 30 studies analyzing EV-derived miRNAs in the context of atherosclerosis have been published, mainly focusing on their role in intercellular communication and their potential use as biomarkers. Some studies have compared the expression of various circulating and EV-derived miRNAs in other diseases such as prostate cancer, concluding that EV-derived miRNAs have greater stability and specificity [49]. To our knowledge, the only study reporting an EV-derived miRNA signature in atherosclerosis was performed in *ldlr* −/− mice [50]. Thus, our study is the first to integrate microarray screening, cross-species bioinformatic analysis, and RT-qPCR validation to identify an EV-derived miRNA signature in individuals with SCA.

Several limitations of our study should be acknowledged. Firstly, the sample size of both the discovery and validation cohorts was small and limited to male participants. In addition, no longitudinal follow-up was conducted to assess the predictive value of the identified signature. Moreover, the cross-sectional design precludes causal inference, and functional validation of predicted targets was not performed. Finally, because we did not analyze differentially expressed circulating miRNAs, the advantage of studying EV-derived miRNAs remains to be confirmed. Future prospective studies should thus aim to validate the predictive value of this miRNA signature in larger, more diverse populations, considering potential sex differences and integrating traditional risk factors and imaging data for preventive cardiology.

In summary, this study provides novel evidence supporting the role of EV-associated miRNAs as early biomarkers and potential regulators of subclinical atherosclerosis. The identified signature may serve as a risk stratification tool and provide insight into the molecular mechanisms underlying asymptomatic vascular disease.

## 4. Materials and Methods

### 4.1. Ethics Statement

This study was conducted according to the principles expressed in the Declaration of Helsinki. Both SCA individuals and controls are Genetics of Atherosclerotic Disease (GEA) cohort participants, unrelated and of self-reported Mexican-mestizo ancestry (3 generations) [51]. Controls were healthy asymptomatic individuals without a family history of premature Coronary Artery Disease (CAD), recruited from blood bank donors at the Instituto Nacional de Cardiología “Ignacio Chávez” (INCICH) in Mexico City and through brochures posted in Social Services centers. Exclusion criteria for controls included congestive heart failure, liver, renal, thyroid, or oncological disease. The research and Ethics Committees of the INCICH and Instituto Nacional de Medicina Genómica (INMEGEN) approved the study protocol. Written informed consent was obtained from all study participants before blood collection.

### 4.2. Subjects

All participants from both cohorts were recruited, and all anthropometric and biochemical measurements were performed at the Department of Endocrinology of the Instituto Nacional de Cardiología Ignacio Chávez (INCICH). All participants underwent computed tomography angiography during a second visit.

*Discovery Cohort.* Eight SCA male subjects and eight male controls from INCICH were enrolled in this study (Table 3). The SCA group included participants with coronary artery calcium score (CACs) ≥ 400 AU, the control group included individuals without evidence of plaque (CACs = 0 AU) matched for age and body mass index (BMI) category (18.5–24.9 kg/m^2^ as healthy weight, ≥25–29.99 kg/m^2^ as overweight and ≥30.0 as obesity, proposed by World Health Organization (WHO).

*Validation Cohort.* The validation cohort included 17 SCA male subjects and 17 male controls recruited at INCICH (Table 4). The validation SCA group included individuals with CACs ≥ 100 AU, and the control group included individuals without evidence of plaque (CACs = 0 AU) matched for age and body mass index (BMI) category. In both cohorts, individuals with an established diagnosis of other diseases such as diabetes mellitus, systemic arterial hypertension, renal or hepatic insufficiency, thyroid disease, acute infections, and with a hereditary family history of CVD or who had presented any clinical manifestation of CAD were excluded from the study.

### 4.3. Sample Collection

After a minimum 12 h overnight fast, blood samples were collected in EDTA tubes, and plasma was obtained by centrifugation at 1500× *g* for 15 min at 4 °C and stored at −80 °C until further processing. In fresh plasma, lipid profile and glucose levels were measured following standardized procedures in the Department of Endocrinology at INCICH.

### 4.4. Discovery Phase

#### 4.4.1. Isolation of EVs

We added 4.5 mL of Phosphate-buffered saline (PBS, pH 7.4,, Sigma-Aldrich, St. Louis, MO, USA) and 2 mL of 50% polyethylene glycol 8000 (PEG 8000, Sigma-Aldrich, St. Louis, MO, USA) to 500 μL of plasma and incubated for 1 h on ice. The mixture was then centrifuged at 1500× *g* for 30 min at 4 °C, and the pellet was resuspended in 1 mL of 0.32% PBS/citrate, pH 7.4. Size exclusion chromatography was performed, and thirty fractions were collected [52]. The eight vesicle-containing fractions were selected according to absorbance at 280 nm and concentrated by ultracentrifugation at 120,000 RFC (53,000 RPM, k-factor 54) at 4 °C for 30 min at 4 °C in a fixed angle rotor (TLA 100.3, Beckman Coulter, Brea, CA, USA). The pellet was resuspended in 60 μL of RIPA buffer with protease inhibitors and EDTA or in 1X PBS, depending on subsequent use.

#### 4.4.2. EV Characterization by Transmission Electron Microscopy (TEM) and Western Blot (WB)

After ultracentrifugation, vesicles were resuspended in 200 μL of PBS. Using Amicon Ultra 3k tubes (Merck Millipore, Burlington, MA, USA), the sample volume was reduced to 100 μL. The EVs were fixed with 400 μL of fixative solution (formaldehyde/glutaraldehyde 2.5%, 0.1 M cacodylate buffer, pH 7.4) and incubated for 45 min at room temperature. The volume was again reduced to 100 μL. Once fixed, seven μL of sample were placed on Formvar/Carbon-coated copper grids and counterstained with 1% alcoholic uranyl acetate for 15 min. Finally, the grids were observed at 60,000× magnification using a JEOL JEM-1010 transmission electron microscope (JEOL Ltd., Tokyo, Japan) equipped with an AMT digital camera.

Total protein in EV preparations were quantified with the Bicinchoninic Acid (BCA) assay following the manufacturer’s instructions. Samples were separated by electrophoresis on 10% polyacrylamide gels using Stain-Free technology (Bio-Rad Laboratories, Hercules, CA, USA), under non-reducing conditions and with a constant protein load (8 µg per sample). Total protein staining using Stain-Free imaging was used as a loading control. Gels were activated with UV light in a ChemidocTM MP Imaging System (BioRad Laboratories, Hercules, CA, USA) according to the manufacturer’s instructions. Immunodetection was performed using anti-CD9 (Ts9 clone, 10626D, Life Technologies, Carlsbad, CA, USA) and anti-CD63 (sc-5275, Santa Cruz Biotechnology, Dallas, TX, USA) antibodies at 1:1000 and 1:2500, respectively, followed by secondary antibody (IgGκ-binding protein-HRP; Santa Cruz Biotechnology, Dallas, TX, USA; sc516102) at 1:2500. Blots were revealed using luminol and bands were visualized with a ChemidocTM MP Imaging System. CD9 and CD63 signals were normalized to total protein and quantified using the ImageLab software, version 6.1 (Bio-Rad Laboratories, Hercules, CA, USA).

#### 4.4.3. Total RNA Isolation

Total RNA, including small RNA species, was extracted from EVs using the exoRNeasy Maxi Kit (QIAGEN, Hilden, Germany), following the manufacturer’s instructions with modifications for a 6 mL plasma volume. The purified RNA was stored at −80 °C until further analysis.

#### 4.4.4. miRNA Expression Profiling

RNA integrity was determined using a 2100 Agilent Bioanalyzer with a Small RNA Chip (Agilent Technologies, Santa Clara, CA, USA); only samples with an integrity score > 8.0 were processed. RNA expression profiles were obtained using the Affymetrix GeneChip miRNA 4.0 Array (Cat. 902411), which includes 30,434 mature miRNA probes, 2578 of which are human miRNAs, and processed at the Microarray Unit of INMEGEN. The microarray CEL format files were processed by the Robust Multiarray Analysis (RMA) method in the R-BiocMananger environment (Bioconductor, version 3.18, Fred Hutchinson Cancer Research Center, Seattle, WA, USA). miRNAs showing −log2FC ≤ 0.5 or ≥0.5, with a *p*-value < 0.05 were considered differentially expressed (DE-miRNAs).

#### 4.4.5. Bioinformatic Analysis of Differentially Expressed miRNAs

##### Selection of Candidate miRNAs

All DE-miRNAs observed in the discovery cohort, which were also found to be differentially expressed in atherosclerotic and control mice Gene Expression Omnibus (GEO) repository of the National Center for Biotechnology Information; https://www.ncbi.nlm.nih.gov/geo/, access number GSE137581, (accessed on 3 June 2023), were selected using R v4.4.3 software and BiocManager packages.

##### Prediction of miRNA Target Genes

Predicted miRNA target genes were identified using TargetScan (https://www.targetscan.org/vert_80/, accessed on 1 August 2025), miRWalk (http://mirwalk.umm.uni-heidelberg.de/, accessed on 1 August 2025), miRmap (https://mirmap.ezlab.org/, accessed on 1 August 2025), and miRDB (https://mirdb.org/, accessed on 1 August 2025) databases. Only target genes identified in at least three databases were considered.

##### Selection of Candidate Genes and Pathway Enrichment Analysis

To ensure that the identified target genes are involved in the human atherosclerosis process, the set of identified genes was compared to genes previously found to be differentially expressed (DEGs) in 68 advanced human atherosclerotic plaques (15 aortic, 29 carotid, and 24 femoral) and 28 controls (left internal thoracic artery [LITA]) from the Tampere Vascular Study [53]. Genes identified by both strategies were analyzed for pathway enrichment using the ShinyGO v0.82 database (https://bioinformatics.sdstate.edu/go/, accessed on 1 August 2025) and the KEGG database (Kyoto Encyclopedia of Genes and Genomes; https://www.kegg.jp/, accessed on 3 June 2023) in *Homo sapiens*. Genes implicated in atherosclerosis-related signaling pathways were then used to generate an interaction network between miRNAs and their respective target genes using Cytoscape v3.7.2 software.

### 4.5. Validation Phase

#### Quantification of miRNA Expression by RT-qPCR

The expression of selected miRNAs was assessed using the following TaqMan^TM^ MicroRNA Assays (Applied Biosystems, Bedford, MA, USA): hsa-miR-146b-5p (Assay ID: 001097, Cat. No.: 4427975), hsa-miR-379-5p (Assay ID: 001138, Cat. No.: 4427975), hsa-miR-487b-3p (Assay ID: 001285, Cat. No.: 4427975), hsa-miR-6849-5p (Assay ID: 466884_mat, Cat. No.: 4440886), hsa-miR-4701-3p (Assay ID: 464888_mat, Cat. No.: 4440886), and hsa-miR-1180-3p (Assay ID: 467239_mat, Cat. No.: 4440886). Reverse transcription was carried out using custom stem-loop primers (Applied Biosystems) designed to specifically recognize the mature miRNA sequences, based on annotations from miRBase (http://www.miRBase.org, accessed on 3 June 2023). The mature sequences of the analyzed miRNAs are provided in Appendix A. Quantitative PCR amplification was conducted on a CFX Opus 96 Real-Time PCR System (Bio-Rad, Hercules, CA, USA).

### 4.6. Statistical Analysis

The Shapiro–Wilk test was used to determine the distribution of the data. In descriptive analysis, quantitative variables were reported as medians and interquartile ranges, and qualitative variables as frequencies and percentages. For inferential analysis, the Mann–Whitney U test was used to compare differences in quantitative variables between groups. Relative expression levels of miRNAs were calculated using the 2^−ΔΔCt^ method, after normalization to the endogenous reference hsa-miR-16. Differences in miRNA expression were also assessed using the Mann–Whitney U test. A *p*-value < 0.05 was considered statistically significant. The Statistical analyses were performed using the Statistical Package for the Social Sciences (SPSS v26, IBM, Armonk, NY, USA). Receiver operating characteristic (ROC) curves and area under the curve (AUC) analyses were generated using GraphPad Prism v8.1 (GraphPad Software, La Jolla, CA, USA) and R software v4.4.3. Bioinformatic analyses of microarray data and pathway enrichment were conducted using R software v4.4.3, along with the BiocManager, limma, affy, and oligo packages (Bioconductor, Fred Hutchinson Cancer Research Center, Seattle, WA, USA), as well as ShinyGO v0.82 and Cytoscape v3.7.2.4.

The diagnostic value of selected miRNAs (individually or combined) was estimated with receiver operating characteristic (ROC) curves using GraphPad Prism v8.1. Relative expression values of miRNAs were combined using the binary logistic tool on SPSS version 26 (IBM Corp., Armonk, NY, USA), where the experimental group was considered the dependent variable, and the selected miRNAs were considered as covariates. The cut-off point was determined using the Youden index, and the value of *p* tested the null hypothesis that the Area under the curve (AUC) equals 0.5.

## 5. Conclusions

This study provides novel evidence supporting the role of plasma EV–derived miRNAs as potential biomarkers of SCA. By integrating microarray profiling, cross-species bioinformatic filtering, and RT-qPCR validation, we identified a six-miRNA (miR-146b-5p, miR-4701-3p, miR-487b-3p, miR-379-5p, miR-1180-3p, and miR-6849-5p) differentially expressed in individuals with SCA. The combination of miR-146b-5p, miR-4701-3p, and miR-1180-3p demonstrated the best clinical diagnostic performance (AUC = 0.8281; 93.75% sensitivity; 93.75% specificity), suggesting potential clinical utility for early diagnosis.

The differential expression of these miRNAs in EVs highlights their relevance, not only as stable circulating biomarkers, but also as possible mediators of intercellular communication in vascular pathology. Their expression patterns may reflect early pathophysiological processes such as endothelial dysfunction, vascular inflammation, lipid dysregulation, and thrombotic activation.

Further research is warranted to validate these findings in larger and more diverse cohorts, including female participants, and to evaluate their longitudinal prognostic value. Mechanistic studies are also needed to characterize the specific targets and functional effects of these miRNAs in vascular cells. Ultimately, the integration of EV-derived miRNA signatures in risk stratification models could improve the identification of individuals at risk for atherosclerosis, allowing early prevention strategies in precision cardiovascular medicine.

## Figures and Tables

**Figure 1 ijms-26-08727-f001:**
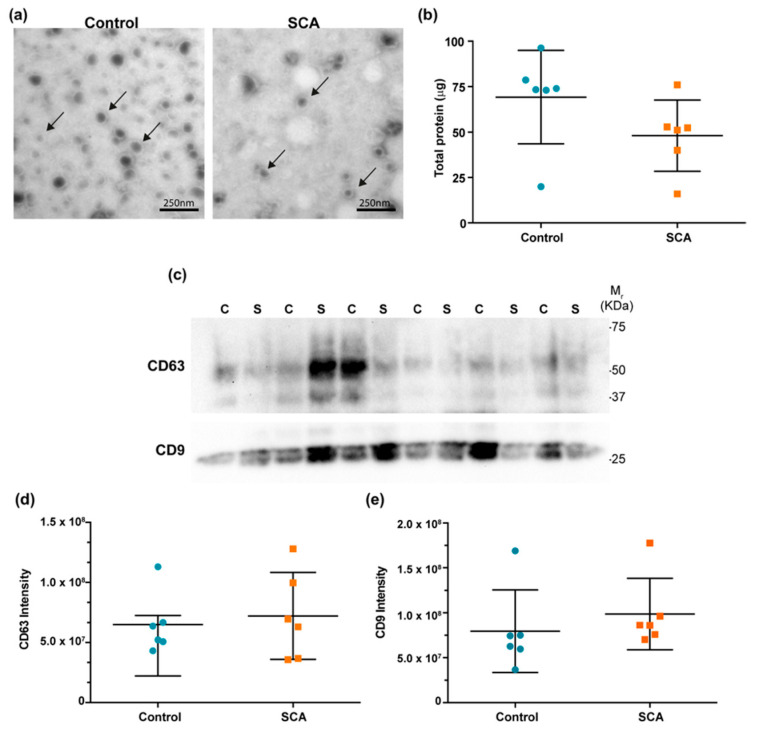
Characterization of plasma-derived (EVs in control and SCA groups. (**a**) Representative transmission electron microscopy (TEM) images showing EV morphology in control and SCA samples (arrows indicate vesicles). Scale bar = 100 nm. (**b**) Total protein concentration (µg) measured in isolated EVs from each group. (**c**) Western blot analysis of canonical EV markers CD63 and CD9 in individual samples (C: control; S: SCA). Molecular weight markers (M_1_) are indicated on the right. (**d**) Densitometric quantification of CD63 band intensity. (**e**) Densitometric quantification of CD9 band intensity. No statistically significant differences were observed between groups. Each dot represents one subject; bars indicate median and interquartile range. Blue circles represent control samples and orange squares represent SCA samples.

**Figure 2 ijms-26-08727-f002:**
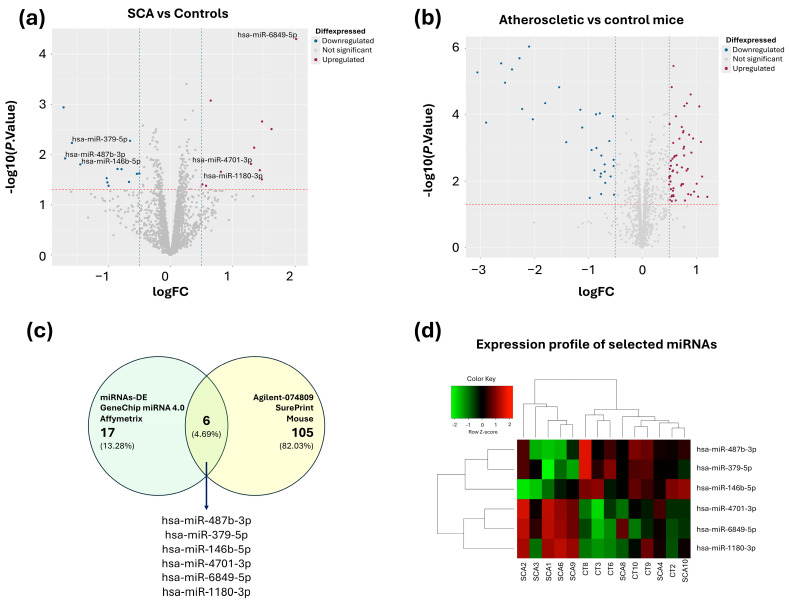
Selection of candidate miRNAs. (**a**) Volcano plot showing the differentially expressed miRNAs (DEG miRNAs) in SCA cases and controls. (**b**) Volcano plot showing DEG miRNAs in mice with and without atherosclerosis, the blue lines represent the cut-off points used to describe fold change on a log2 scale (Fold change < −0.5 and >0.5), while the red line represents the *p*-value threshold on a log10 scale (<0.05). (**c**) Venn diagram showing miRNAs differentially expressed both in mice and humans with atherosclerosis. (**d**) Heatmap showing the expression profiles of the selected miRNAs.

**Figure 3 ijms-26-08727-f003:**
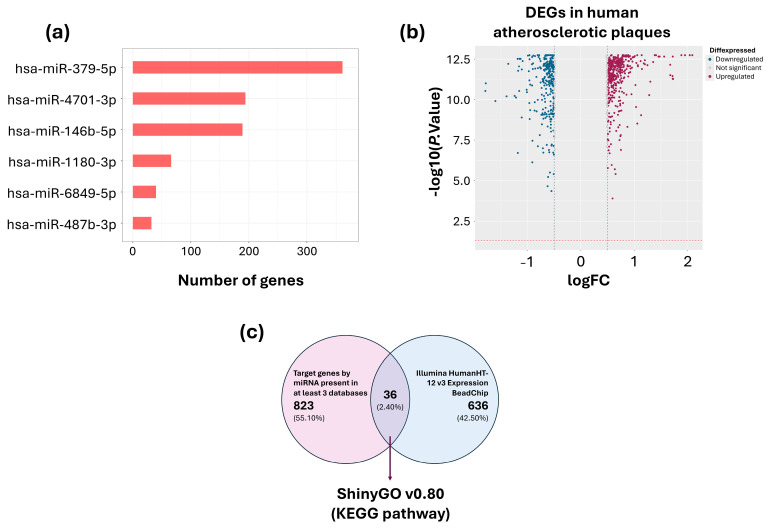
Selection of target genes. (**a**) Number of target genes of each selected miRNA reported in at least three databases. (**b**) Volcano plot of the differentially expressed genes (DEGs) in human atherosclerotic plaques The blue lines represent the cut-off points used to describe the rate of change on a log2 scale (Fold change < −0.5 and >0.5), while the red line represents the *p*-value on a log10 scale (<0.05). (**c**) Venn diagram showing target genes identified by both approaches.

**Figure 4 ijms-26-08727-f004:**
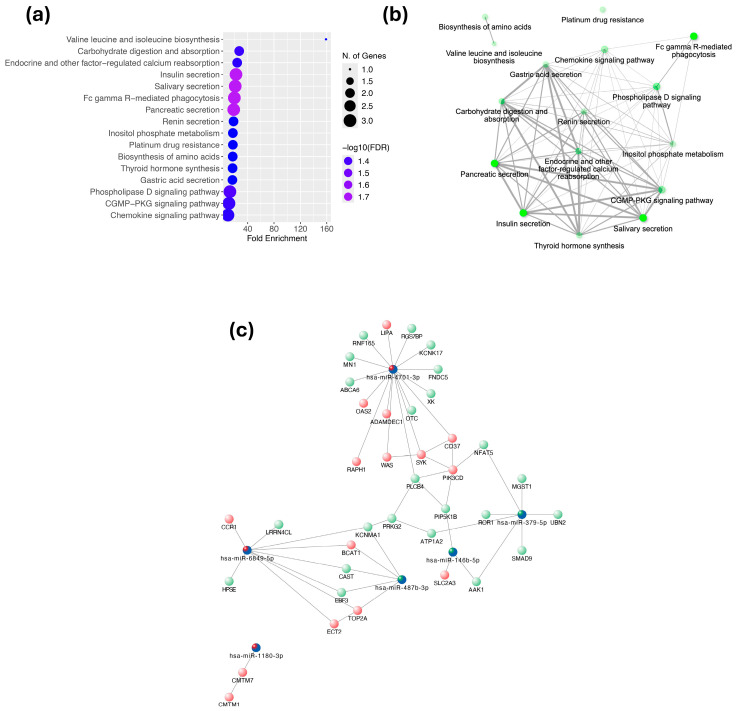
Pathway enrichment analysis and interaction networks. (**a**) Signaling pathway enrichment analysis for selected target genes. (**b**) Interaction network between the selected signaling pathways. (**c**) Interaction network between miRNAs and their potential target genes. Downregulated genes are shown in green, and upregulated genes are shown in red. The six candidate miRNAs are highlighted in dark blue, while other downregulated miRNAs are shown in light blue.

**Figure 5 ijms-26-08727-f005:**
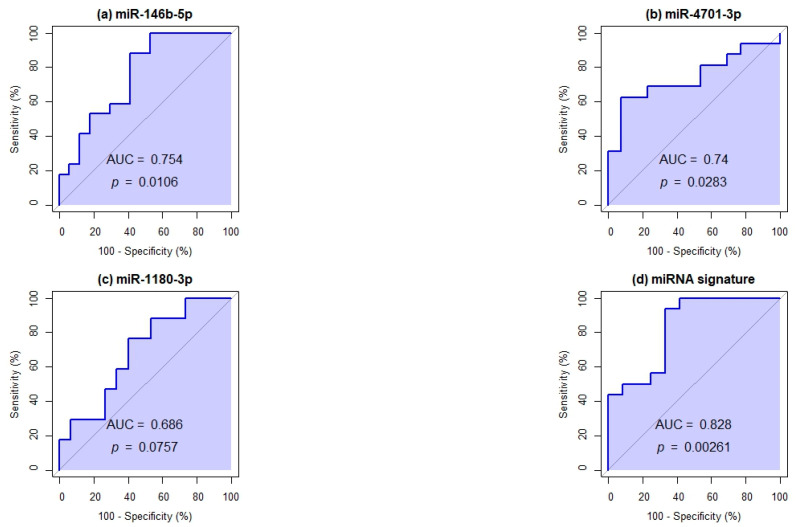
ROC curve analysis of individual and combined miRNAs. Panels (**a**–**c**) show the individual diagnostic performance of miR-146b-5p, miR-4701-3p, and miR-1180-3p, respectively. Panel (**d**) depicts the combined three-miRNA signature, which achieved the best classification performance. The AUC and *p*-values are indicated in each graph.

**Table 1 ijms-26-08727-t001:** Relative expression of selected plasma EV-derived miRNAs in individuals with SCA and controls.

miRNA	Control (*n* = 17)	SCA (*n* = 17)	*p*
miR-146b-5p	14.04 (5.850–25.663)	4.928 (1.806–16.079)	0.006
miR-4701-3p	0.054 (0.025–0.077)	0.083 (0.059–0.108)	0.014
miR-487b-3p	1.942 (1.497–2.094)	0.526 (0.274–1.573)	0.034
miR-379-5p	1.341 (0.807–1.424)	3.260 (2.109–8.056)	0.036
miR-1180-3p	0.028 (0.002–0.045)	0.084 (0.046–0.200)	0.036
miR-6849-5p	0.007 (0.006–0.017)	0.012 (0.006–0.067)	0.048

Values are presented as medians and interquartile ranges (IQR). Comparisons between groups were performed using the Mann–Whitney U test. miRNA = microRNA

**Table 2 ijms-26-08727-t002:** ROC curve analysis showing the diagnostic performance of selected circulating miRNAs, evaluated individually and in combination, to discriminate SCA from controls.

miRNA	AUC	*p*	Sensitivity%	Specificity%
miR-146b-5p	0.7543	0.0106	88.24	58.82
miR-4701-3p	0.7404	0.0283	62.5	92.31
miR-487b-3p	0.6902	0.067	66.67	76.47
miR-379-5p	0.6863	0.0729	86.67	52.94
miR-1180-3p	0.6863	0.0757	76.47	60
miR-6849-5p	0.6765	0.0955	41.18	100
miR-146b-5p + miR-4701-3p	0.8413	0.0018	87.5	69.23
miR-146b-5p + miR-487b-3p	0.8693	0.0013	93.75	63.64
miR-146b-5p + miR-4701-3p + miR-487b-3p	0.8846	0.0007	100	69.23
miR-146b-5p + miR-4701-3p + miR-379-5p	0.8791	0.0008	100	69.23
miR-146b-5p + miR-4701-3p + miR-1180-5p	0.8281	0.0026	93.75	93.75
miR-146b-5p + miR-4701-3p + miR-487b-3p + miR-379-5p	0.8791	0.0008	100	69.23
miR-146b-5p + miR-4701-3p + miR-487b-3p + miR-1180-3p	0.869	0.0014	100	66.67

Values represent AUC, sensitivity, and specificity from ROC analysis. The miRNA combination was determined by logistic regression. Statistical significance was assessed using the Mann–Whitney U test (*p* < 0.05). AUC = area under the curve; miRNA = microRNA; CI = confidence interval.

**Table 3 ijms-26-08727-t003:** Main clinical and biochemical characteristics of participants in the discovery phase.

	Control (*n* = 8)	SCA (*n* = 8)	*p*
Age (years)	66.5 (63.75–69.5)	68.5 (66–72.5)	0.594
Weight (kg)	71.35 (67.425–86.45)	80.25 (65.05–95.15)	0.529
Height (m)	1.655 (1.575–1.715)	1.655 (1.640–1.772)	0.526
BMI (kg/m^2^)	28.23 (24.707–30.45)	29.61 (24.00–30.75)	0.875
Glucose (mg/dL)	101.8 (90.725–113.6)	96.30 (91.15–103.1)	0.529
Total cholesterol (mg/dL)	162.2 (144.15–176.4)	165.7 (110.4–197.5)	1.000
HDL-c (mg/dL)	40.20 (32.150–45.07)	40.35 (30.70–45.15)	1.000
LDL-c (mg/dL)	92.80 (69.650–109.9)	113.3 (50.75–126.4)	0.462
Triglycerides (mg/dL)	133.8 (72.175–216.1)	165.1 (119.4–296.7)	0.401
CACS (AU)	0.0 (0.0–0.0)	483.4 (436.9–770.1)	<0.0001

Values are presented as medians and interquartile ranges (IQR). Comparisons were performed using the Mann–Whitney U test. SCA = Subclinical coronary atherosclerosis; CACS = coronary artery calcium score; BMI = body mass index; HDL-c = high-density lipoprotein cholesterol; LDL-c = low-density lipoprotein cholesterol.

**Table 4 ijms-26-08727-t004:** Demographic, anthropometric, and biochemical characteristics of the validation cohort (SCA cases and controls).

	Control (*n* = 17)	CSA (*n* = 17)	*p*
Age (years)	67.0 (57.0–72.5)	67.0 (61.0–72.0)	0.666
Weight (kg)	77.0 (67.45–81.8)	77.6 (72.15–88.90)	0.491
Height (m)	1.67 (1.635–1.719)	1.63 (1.60–1.73)	0.73
BMI (kg/m^2^)	28.50 (23.98–29.86)	28.97 (27.16–30.27)	0.38
Body fat (%)	753.0 (432.0–1315.0)	654.0 (442.0–1121.0)	0.796
Glucose (mg/dL)	95.8 (90.8–106.0)	97.1 (89.5–113.2)	0.931
Total cholesterol (mg/dL)	185.7 (165.2–205.1)	176.0 (151.8–196.4)	0.418
HDL-c (mg/dL)	38.8 (34.8–47.9)	44.8 (35.4–50.9)	0.558
LDL-c (mg/dL)	117.0 (101.7–131.4)	111.3 (87.0–130.3)	0.667
Triglycerides (mg/dL)	149.8 (117.4–201.9)	130.8 (103.2–198.5)	0.547
CACS (AU)	0.0 (0.0–0.0)	200.5 (118.2–293.9)	<0.0001

Values are presented as medians and interquartile ranges (IQR). Comparisons were performed using the Mann–Whitney U test. BMI = body mass index. CACS = coronary artery calcium score. HDL-C = high-density lipoprotein cholesterol. LDL-C = low-density lipoprotein cholesterol. CT = total cholesterol. TG = triglycerides.

## Data Availability

The data supporting the findings of this study are available from the corresponding author upon reasonable request.

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
