# Peer review of "A Plasma Extracellular Vesicle-Derived microRNA Signature as a Potential Biomarker for Subclinical Coronary Atherosclerosis"

_ijms, 2025, doi:10.3390/ijms26178727_

Round 1

Reviewer 1 Report

Comments and Suggestions for Authors

In the manuscript, the authors focused on profiling the possible miRNA biomarkers of subclinical coronary atherosclerosis (SCA) in plasma extracellular vesicle (EV) using clinical sample analysis and bioinformatics. And the auhtors successfully found 6 especially 3 miRNAs in plasma EV which present high potential as SCA biomarkers. Just as mentioned according to the authors, this manuscript integrated microassay screening, cross-species bioinformatic analysis, and RT-PCR validation to identify the EV-drived miRNA signature in individuals withe SCA. The creativity and the workload of the manuscript are enough. And the manuscript may expand the border of non invasive disease diagnosis and provide a good model and strategy for personal therapy and disgnosis. The manuscript is recommended to be accepted by the International Journal of Molecular Sciences. Here are some concerns for the authors to improve the quality of the manuscript:

1) In the Keywords section, words like RT-qPCR, inflammation, lipid metabolism, vascular remodeling should be deleted.

2) In the 7th paragraph of the Introduction section, in the 1st sentence, the auhtors mentioned the background of miRNA which is  in accordance with Prof. Cheng's previous work (10.1021/am500883q). Therefore, his work is recommended to be cited but it is not in necessity.

3) In section 2.3, the authors obtained the plasma through centrifugation, the authors should provide the detail parameters including temperature and speed.

4) In section 2.4.1, the authors should provide the pH value of the PBS in the 1st sentence.

5) In the 2nd sentence of section 2.4.2, the sentence should be revised using the research subject as the subject. And the authors should provide the origin of the TEM.

6) In section 2.5, the authors should provide the detail primer sequences used in the manuscript to the SPI file.

7) In section 3.1.2, the authors should revise "Figure 1A (or B, C, D, E)" as "Figure 1a (or b, c, d, e)".

8) Figure 3d and Figrue 3e were not read in the manuscript.

9) Here is an interesting question: The authors got high AUC for each individual screened out miRNA, have the authors considered infuge these miRNA together to improve the AUC for SCA diagnosis?

Overall, this is a good work. I provide minor revision recommendation. However, I want to hear the authors' reply about the 9th concern.

Author Response

We sincerely thank you for taking the time to review the manuscript and for your valuable observations.

1) In the Keywords section, words like RT-qPCR, inflammation, lipid metabolism, vascular remodeling should be deleted.

According to your observation, we eliminated the terms “RT-qPCR”, “inflammation”, “lipid metabolism”, and “vascular remodeling” from the Keywords list.

2) In the 7th paragraph of the Introduction section, in the 1st sentence, the auhtors mentioned the background of miRNA which is  in accordance with Prof. Cheng's previous work (10.1021/am500883q). Therefore, his work is recommended to be cited but it is not in necessity.

We added the recommended reference (Cheng et al., 2014; now reference 27) in the first sentence of the seventh paragraph of the Introduction to acknowledge this previous work.

3) In section 2.3, the authors obtained the plasma through centrifugation, the authors should provide the detail parameters including temperature and speed.

We added the centrifugation parameters for plasma separation in Section 2.3, indicating that blood samples were centrifuged at 1500 × g for 15 minutes at 4 °C.

4) In section 2.4.1, the authors should provide the pH value of the PBS in the 1st sentence.

Thank you for your observation. We specified the pH of the PBS (pH 7.4) used for the isolation procedure in Section 2.4.

5) In the 2nd sentence of section 2.4.2, the sentence should be revised using the research subject as the subject. And the authors should provide the origin of the TEM.

We revised the second sentence in Section 2.4.2 to clarify that EVs are the subject of the sentence. In addition, we specified the details of the transmission electron microscope, indicating that the grids were observed at 60,000 × magnification using a JEOL JEM-1010 transmission electron microscope equipped with an AMT digital camera.

6) In section 2.5, the authors should provide the detail primer sequences used in the manuscript to the SPI file.

The exact sequences of the primers are proprietary to Applied Biosystems, as TaqMan™ MicroRNA Assays were used in this study. To ensure transparency and reproducibility, we included the mature miRNA sequences (obtained from miRBase) targeted by these assays in Supplementary Table S5, which is now provided in the Supplementary Information file.

7) In section 3.1.2, the authors should revise "Figure 1A (or B, C, D, E)" as "Figure 1a (or b, c, d, e)".

We revised the references to the subpanels of Figure 1, changing the letters (a-e) from uppercase to lowercase in Section 3.1.2 and in the corresponding figure legend.

8) Figure 3d and Figrue 3e were not read in the manuscript.

We have removed the descriptions of panels (d) and (e) from the Results section. The revised text now only refers to panels (a–c), which correspond to the final version of Figure 3.

9) Here is an interesting question: The authors got high AUC for each individual screened out miRNA, have the authors considered infuge these miRNA together to improve the AUC for SCA diagnosis?

Thank you for this valuable comment. We assessed the diagnostic performance of combined miRNAs using multivariable logistic regression followed by ROC curve analysis. As shown in Table 4 and Figure 5, the combination of miR-146b-5p, miR-4701-3p, and miR-1180-3p achieved the best classification performance, with an AUC of 0.8281, 93.75% sensitivity, and 93.75% specificity. We clarified this point in the revised version of the manuscript to emphasize that combining selected miRNAs improved diagnostic accuracy as compared to individual markers.

Overall, this is a good work. I provide minor revision recommendation. However, I want to hear the authors' reply about the 9th concern.

Reviewer 2 Report

Comments and Suggestions for Authors

The study of early diagnosis of cardiovascular diseases is an extremely important task for the entire world. The work is done and described carefully, there are practically no comments. Only a few questions and suggestions arose:

Line 136. It is said that there were 8 men in the SCA of Discovery Cohort group, but it is not clear what gender the control group was.

Line 280. An extra opening bracket.

Line 293. It is not clear why late stages of the disease were compared with a mouse model. Undoubtedly, mouse models are very interesting and useful, but the question is: Are data on people with a late stage unavailable?

Line 324. The explanation to Figure 4a says "signaling pathways", but this is probably a typo, since 4a indicates not only signaling pathways. Another question that readers will have about "N. of Genes" in this figure is that if this is the number of genes, how can it not be an integer (1.5, 2.5)?

Line 334. The table title lacks an explanation that it is about a validation cohort, otherwise the difference from Table 1 is unclear.

Line 508. The names of the authors of publication #1 are missing.

The entire list of references is formatted with extra commas between last names and initials, and extra ";" instead of commas.

Author Response

In response to the observations of Reviewer 2:

Thank you sincerely for taking the time to review our manuscript and for your valuable observations.

The study of early diagnosis of cardiovascular diseases is an extremely important task for the entire world. The work is done and described carefully, there are practically no comments. Only a few questions and suggestions arose:

Line 136. It is said that there were 8 men in the SCA of Discovery Cohort group, but it is not clear what gender the control group was.

We clarified that all participants in the discovery cohort, including both SCA and control groups, were male. This correction was made in the second paragraph of section “2.2 Subjects”

Line 280. An extra opening bracket.

We revised the formatting and added the missing space between the number and the parenthesis to improve clarity. This correction was made in section 4.1.1 Main Clinical Features of Study Participants (Table 1).

Line 293. It is not clear why late stages of the disease were compared with a mouse model. Undoubtedly, mouse models are very interesting and useful, but the question is: Are data on people with a late stage unavailable?

To narrow down the universe of miRNAs from our microarray data implicated in atherosclerosis, we conducted a thorough search in various information repositories using keywords such as "miRNA", "microRNAs", "Atherosclerosis", "Differential expression", "microarrays", and "seqRNA". The only publication that met these parameters and provided data available for analysis was reported by Guo J et al. 2020 (accession number GSE137581).  We agree that comparative analyses should use data that meet very similar conditions. The study of Guo et al. provided the data most similar to ours.  We did not find data on late-stage atherosclerosis in humans available for analysis.  

Line 324. The explanation to Figure 4a says "signaling pathways", but this is probably a typo, since 4a indicates not only signaling pathways. Another question that readers will have about "N. of Genes" in this figure is that if this is the number of genes, how can it not be an integer (1.5, 2.5)?

We modified the text in section 3.1.4 to clarify that Figure 4a shows the 16 signaling pathways, the number of genes, and the –log10(FDR) values. The values of the “Number of Genes” parameter are generated directly by the software, which scales circle sizes proportionally to the number of genes associated with each pathway. This procedure may yield fractional values (e.g., 1.5 or 2.5) as an approximation, and it can be reproduced with the Chart function in ShinyGO v0.82.

Line 334. The table title lacks an explanation that it is about a validation cohort, otherwise the difference from Table 1 is unclear.

We modified the title of Table 2 to clarify that it corresponds to the validation cohort. Table 1 describes the characteristics of the discovery cohort. This correction was made in section 4.2.1 Main Clinical Features of Study Participants.

Line 508. The names of the authors of publication #1 are missing.

Thank you for your careful review. We have completed the reference by adding the missing authors’ names.

The entire list of references is formatted with extra commas between last names and initials, and extra ";" instead of commas.

We revised the reference list carefully. The format used in our manuscript follows the International Journal of Molecular Sciences (IJMS) guidelines, which specify commas between last names and initials, and semicolons to separate authors, as shown in the journal’s reference template. We corrected minor issues such as duplicated years or inconsistent volume formatting to ensure full compliance with the IJMS style.